# Intranasal Immunization for Zika in a Pre-Clinical Model

**DOI:** 10.3390/v16060865

**Published:** 2024-05-28

**Authors:** Sarthak Shah, Parth Patel, Priyal Bagwe, Akanksha Kale, Amarae Ferguson, Emmanuel Adediran, Tanisha Arte, Revanth Singh, Mohammad N. Uddin, Martin J. D’Souza

**Affiliations:** Vaccine Nanotechnology Laboratory, Center for Drug Delivery Research, College of Pharmacy, Mercer University, Atlanta, GA 30341, USA

**Keywords:** Zika, intranasal, vaccine, microparticles, Guillain–Barré syndrome, microcephaly

## Abstract

Humans continue to be at risk from the Zika virus. Although there have been significant research advancements regarding Zika, the absence of a vaccine or approved treatment poses further challenges for healthcare providers. In this study, we developed a microparticulate Zika vaccine using an inactivated whole Zika virus as the antigen that can be administered pain-free via intranasal (IN) immunization. These microparticles (MP) were formulated using a double emulsion method developed by our lab. We explored a prime dose and two-booster-dose vaccination strategy using MPL-A^®^ and Alhydrogel^®^ as adjuvants to further stimulate the immune response. MPL-A^®^ induces a Th1-mediated immune response and Alhydrogel^®^ (alum) induces a Th2-mediated immune response. There was a high recovery yield of MPs, less than 5 µm in size, and particle charge of −19.42 ± 0.66 mV. IN immunization of Zika MP vaccine and the adjuvanted Zika MP vaccine showed a robust humoral response as indicated by several antibodies (IgA, IgM, and IgG) and several IgG subtypes (IgG1, IgG2a, and IgG3). Vaccine MP elicited a balance Th1- and Th2-mediated immune response. Immune organs, such as the spleen and lymph nodes, exhibited a significant increase in CD4^+^ helper and CD8^+^ cytotoxic T-cell cellular response in both vaccine groups. Zika MP vaccine and adjuvanted Zika MP vaccine displayed a robust memory response (CD27 and CD45R) in the spleen and lymph nodes. Adjuvanted vaccine-induced higher Zika-specific intracellular cytokines than the unadjuvanted vaccine. Our results suggest that more than one dose or multiple doses may be necessary to achieve necessary immunological responses. Compared to unvaccinated mice, the Zika vaccine MP and adjuvanted MP vaccine when administered via intranasal route demonstrated robust humoral, cellular, and memory responses. In this pre-clinical study, we established a pain-free microparticulate Zika vaccine that produced a significant immune response when administered intranasally.

## 1. Introduction

The Zika virus (ZIKV) is mosquito-borne and is a member of the *Flaviviridae* family. This virus was originally named after a forest in Uganda in 1947 [1]. In 1952, the first human case was documented in the United Republic of Tanzania and Uganda. After the discovery of the first case, an outbreak of ZIKV occurred in 2007 in Micronesia and then again during 2013–2014 in New Caledonia and Polynesia. To date, more than 80 countries have reported the presence of the Zika virus [2].

Once an individual is infected with the Zika virus, several symptoms may develop. Symptoms induce headaches, myalgia (muscle weakness), arthralgia (joint pain), and fever [1,3]. Other symptoms may develop such as rash, non-purulent conjunctivitis, and general malaise [4]. An individual can be exposed to this virus via blood transfusion or at a laboratory, academic, or industry setting. New findings or reports detailing the consequences of the Zika infection are increasing every day. Researchers can study the negative consequences that occur based on the trimester of the pregnancy. In the first trimester, a Zika infection causes congenital defects which include microencephaly. However, in the trimester, the Zika infection causes delays in brain development. Recently, it was shown that a congenital Zika infection may trigger hormonal imbalances of the hypothalamic-pituitary–gonadal axis. This imbalance in the hypothalamus can lead to reduced fertility [5]. In adults, Zika infection has been associated with Guillain–Barré syndrome. This syndrome is an autoimmune syndrome that affects the peripheral nervous system (PNS) resulting in paralysis and sometimes death. The current process of managing this infection is supportive; sufficient hydration, antipyretic medication, and analgesics [6].

The Ebola epidemic, in 2014, showed the World Health Organization (WHO) and governments around the world that we were not well equipped [7]. Not wanting to repeat the same mistake, the WHO acted swiftly to raise the alarm [8]. Due to the frequent outbreaks of Zika, the World Health Organization (WHO) declared that this virus be considered a public health concern in 2016 [1]. Particularly, the increased incidence of microcephaly was alarming since this complication was a heart-breaking burden not only on families, but on communities as well. The advantage of a global approach allow for a well-coordinated worldwide effort which will allow for surveillance for microcephaly and for research efforts to be intensified.

There are several ways the Zika virus can be transmitted other than a bite from mosquitoes, specifically Aedes aegypti. Other methods of transmission include congenital, sexual, and intranasal transmission. Clinicians have reported an array of symptoms, including speech delay, intellectual disability, movement issues, and vision and hearing loss [9]. Due to these congenital clinical findings, a new term was coined: congenital Zika syndrome (CZS). Recent studies have shown the transmission of ZIKV via sexual contact [10]. This was due to ZIKV being detected in various body fluids such as saliva, urine, breast milk, semen, and in nasopharyngeal mucosa. Investigators have reported the transmission of ZIKV via oronasal sections in several animal models such as pigs, geese, and ducks [10]. Interestingly, a recent case reported an individual with ZIKV infection; however, there was no known cause such as a bite from mosquito or sexual contact [11]. These findings suggest that close contact with another individual or exposure to the body fluids which have the presence of ZIKV could cause infection. These exciting findings encouraged us to investigate the feasibility of an intranasal vaccine for Zika. An intranasal vaccination approach can help to stimulate the immune system to produce an efficient humoral and cellular response (Figure 1). The nasal cavity is a multicompartment network consisting of the vestibular, respiratory, and olfactory systems [12]. The nasal mucosal defense system consists of inductive and effector sites. An inductive site is the where the immune system initially interacts with the pathogen. These sites are called nasal-associated lymphoid tissues (NALT) and mucosa-associated lymphoid tissues (MALT). These sites have epithelial cells and microfold (M) cells. Inside the pockets of M cells (Figure 2) or the basal site of the M cells, is a rich area consisting of antigen presenting cells (APCs), B cells, T cells, and macrophages for initiating an immune response to the pathogen [13]. Administration of a microparticulate vaccine approach has shown strong immune response for triggering an effective immune response and a robust adaptive response [1,14,15]. Once the vaccine microparticles are administered, the APCs can take up these particles. However, they are not limited to a certain inductive site. Several APCs can migrate to other mucosal sites. The advantage here is that proximal and distant mucosal zones can be used to initiate the immune response to a specific antigen. Once the APCs have been activated, they can migrate to secondary lymphoid organs where follicular B cell cells and interfollicular T cells are housed [16]. Once the immune cells in the secondary lymphoid organs have been activated, they can further activate the immune system. Some of these immune cells can migrate to other areas of the body via the bloodstream. This can not only activate a mucosal but also a systemic response, and thus promotes the production of IgA and IgG. The advantage of IgA production, via the intranasal vaccination route, is that protection is not limited to local nasal and adjacent oral mucosa areas. IgA secretions occur at distant vaginal and rectal surfaces, providing widespread mucosal immunity [10].

To date, there is no approved treatment or vaccine available which raises the possibility of a future reemergence of a viral infection. In our approach, we developed a polymeric microparticulate (MP) vaccine by encapsulating a purified inactivated ZIKV as the antigen. Encapsulating the antigen or adjuvant in microparticulate form offers an advantage over the traditional vaccines: the particle’s appearance seems more unfamiliar or foreign to the immune system, thereby enhancing the initiation of a strong immune response. The application of microparticles in vaccine formulation has demonstrated specific enhancements in antigen stability, availability, adjuvanticity, immunostimulatory capacity, targeted delivery, and precise or sustained release of the antigen [17]. Numerous literatures have demonstrated the effectiveness of using a particulate vaccine over the traditional vaccine [17,18]. These particulate biodegradable carriers represent efficient antigen delivery systems capable of improving the uptake of antigens by antigen-presenting cells (APCs) like dendritic cells (DCs) or macrophages. We combined our vaccination strategy by adding two FDA-approved adjuvants, since many clinical trials have reported enhanced immunity with use of an adjuvant [19]. Adjuvants are substances that boost the stability and immunogenicity of vaccine antigens, adjust efficacy, and elevate immune responses to the antigen [20]. In our study, we investigated our ZIKV MP vaccine with Alhydrogel^®^ (alum) and Monophosphoryl Lipid-A (MPL-A^®^) adjuvants in microparticulate form. Alhydrogel^®^ (alum) helps to stimulate a T-cell helper Th2 antibody-mediated response [21]. On the other hand, MPL-A^®^ helps to induce a Th1 T-cell helper response. This approach will allow for a balanced Th1/Th2 response which is needed for a healthy immune system. This vaccine approach was investigated in the Swiss Webster mice model to determine the vaccine efficiency and correlates of an immune response when administered via the intranasal route.

## 2. Materials and Methods

### 2.1. Materials

The Zika virus strain PRVABC59 was supplied by the Centers for Disease Control and Prevention (CDC), Colorado (viral titer of 1 × 10^6^ PFU/mL). The VERO C1008 cells were bought from American Type Culture Collection (ATCC CRL-1586™) (Manassas, VA, USA). The polymer poly (D, L-lactide-co-glycolide) grade 75:25 (PLGA) (Resomer^®^ RG 752 H) was purchased from Evonik Corporation (Birmingham, AL, USA). In this study, we used two FDA-approved adjuvants, Alhydrogel^®^ and Monophosphoryl Lipid A (MPL-A^®^), that were purchased from InvivoGen (San Diego, CA, USA). Trehalose dihydrate (cryoprotectant) and polyvinyl alcohol (PVA) (grade: Avg Mol Wt. 30,000–70,000) were purchased from Sigma-Aldrich (St. Louis, MO, USA). We used Swiss Webster mice (4–6 weeks old, female) for in vivo studies that were purchased from Charles River Laboratories (Wilmington, MA, USA). For the ELIZA experiments, we purchased Goat anti-mouse secondary IgG, IgG1, IgG2a, IgG3, IgA, and IgM conjugated to Horseradish Peroxidase (HRP) from Invitrogen™ (Rockford, IL, USA). For flow cytometry analysis, Allophycocyanin (APC)-labeled anti-mouse CD4 antibody and fluorescein isothiocyanate (FITC)-labeled anti-mouse CD8a antibody were purchased from Invitrogen™, Thermo Fisher Scientific (Waltham, MA, USA).

### 2.2. Zika Viral Culture and Inactivation

Replication of the Zika virus was conducted using VERO cells. We used a 98–100% confluent monolayer of VERO cells for infection. The T-75 flask was infected with Zika virus at a 0.01 multiplicity of infection (MOI). The supernatant was collected after the cytopathic effect was observed after 1 week or 7 days. After confirming the cytopathic effect, we proceeded with inactivation of the virus where the supernatant was treated with 0.03% beta-propiolactone (BPL) (Fischer Scientific, Hampton, NH) for 72 h to inactivate the virus. After the 72 h, we confirmed the virus inactivation by a cell infectivity assay for three passages in VERO cells. After confirming that the virus was inactivated, the virus was then purified using Centricon^®^ Plus-70 centrifugal filters (Millipore Corporation, Burlington, MA, USA). After centrifugation, we obtained the final concentrate, which was the inactivated purified Zika virus. Using the final concentrate, the protein content was quantified using the Pierce™ bicinchoninic acid (BCA) protein assay kit (Thermo Fischer Scientific, Rockford, IL, USA). This purified, inactivated virus was used as the vaccine antigen for in-vitro and in-vivo studies.

### 2.3. Formulation of Microparticulate Vaccine

The formulation of vaccine microparticles has been published by our group previously [1]. The double emulsion solvent evaporation method was utilized for encapsulating the inactivated Zika virus into the poly (D, L-lactide-co-glycolide) (PLGA) polymer to form MPs (Figure 3). Briefly, the Zika antigen was emulsified in a 2% polymer solution (PLGA in dichloromethane (DCM)) using Omni TH_Q_ probe homogenizer (Kennesaw, GA, USA). This is the primary emulsion. Next, the first emulsion was emulsified with 0.1% polyvinyl alcohol (PVA, MW 30,000–70,000, Sigma-Aldrich) solution. This is the second or double emulsion. To reduce the size, the resultant emulsion was passed through a Nano DeBEE high-pressure homogenizer for six cycles. After passing through the homogenizer, the emulsion was under constant stirring for 4 h to evaporate the DCM. After the evaporation step, the emulsion was placed into a ultracentrifuge machine to concentrate the MP. After ultracentrifuge, this concentrate was then freeze-dried with trehalose as the cryoprotectant using Labconco™ FreeZone Triad benchtop freeze dryer. Two adjuvant MPs encapsulating Alhydrogel^®^ and MPL-A^®^ were formulated using the similar method as mentioned above.

### 2.4. Measurement of Microparticle Recovery Yield, Particle Size, and Zeta Potential

The percentage of recovery yield of lyophilized product was obtained using the following formula: percent recovery yield = (weight of microparticles × 100)/total weight of the formulation [1,18]. The microparticles were carefully weighed with an electronic weight scale. The weight used is the percent recovery yield calculation. The particle size was determined using a 12 mm square polystyrene cuvette (DTS0012). The zeta potential was measured using a folded capillary cell (DTS1070). Particle size and zeta potential was measured using Zetasizer Nano ZS (Malvern Pananalytical, Westborough, MA, USA). The sample was a uniform suspension of MPs with a concentration of 0.05 mg/mL. This experiment was repeated thrice.

### 2.5. Laser Particle Counter

A laser particle counter (Spectrex PC-2200 (Redwood City, CA, USA)) was utilized to determine the number of particles of Zika MPs that are present in 1 mL of PBS [18]. A total of 2 mg of Zika MPs was carefully weighed and placed in 2 mL of phosphate-buffered saline (PBS). The measurements of this experiment were repeated six times (*n* = 6). Adjuvant MP (Alhydrogel and MPL-A) were also measured in the same manner.

### 2.6. Fourier Transform Infrared Microscopy (FTIR)

MPs were analyzed using Fourier Transform Infrared Microscopy (FTIR) (Shimadzu IRAffinity-1S (Tampa, FL, USA)). Briefly, 1 mg of Zika MP PLGA polymer were placed on the ZnSe crystal puck and analyzed. The Zika solution (20 µL) was placed on the ZnSe crystal puck and analyzed.

### 2.7. Evaluation of In Vitro Immunogenicity via Griess Assay

The in vitro immunostimulatory potential of Zika vaccine microparticles with or without adjuvant (alum and MPL-A) microparticles was evaluated by measuring nitric oxide released by dendritic cells DC 2.4 using a Griess assay [1,18]. After the stimulation of APCs with an antigen, nitric oxide, an important innate immunity marker, is released and can be quantified by this assay [1,21]. Briefly, cells were plated in a 96 well plate with a seeding density of 1 × 10^4^ cells/well. The DC cells in different wells were exposed to various treatment groups. Treatment groups: Unstimulated cells were used as control, lipopolysaccharide (LPS) was used as a positive control, inactivated Zika solution (50 µg/well), Zika MP vaccine (50 µg/well), Zika MP (50 µg/well) + adjuvants (25 µg/well), alum MP (50 µg/well), and MPL-A (50 µg/well). After addition of the various treatment groups, cells were placed in an incubator for 24 h. After 24 h, the culture supernatants were collected from each treatment group and treated with the Griess reagent system (1% sulfanilamide in 5% phosphoric acid and 0.1% N-1-naphthyl ethylenediamine dihydrochloride (NED)) (Fischer Scientific, Hampton, NH, USA). This is needed to convert the nitric oxide into nitrite and later form a pink-colored azo compound. After the addition of the Griess reagent system, the 96 well plate was placed in an incubator for 3 h. After the 3 h, the plate was placed into a BioTek^®^ Synergy H1 microplate reader (BIO-TEK Instruments, Winooski, VT, USA) where the absorbance was measured at 540 nm. Next, the concentration of nitrite was calculated using a standard curve of sodium nitrite (Fischer Scientific, Hampton, NH, USA) with the standard concentrations ranging from 3 µM to 200 µM.

### 2.8. Evaluation of In Vitro Cytotoxicity of Zika Vaccine by MTT Assay

The cytotoxicity profile of our vaccine microparticles was established in vitro [1,18]. The cytotoxicity of the ZIKV MP vaccine and adjuvant (alum and MPL-A) MP was investigated by a 3 (4,5-dimethylthiazol-2-yl)-2,5-diphenyl tetrazolium bromide (MTT) cell viability assay [1,18]. Briefly, dendritic 2.4 cells (DC) were plated in a 96 well plate with a seeding density of 1 × 10^4^ cells/well. The cells only group and dimethyl sulfoxide (DMSO) group served as negative and positive controls. DC cells were treated with Zika solution (50 µg/well), Zika MP vaccine (50 µg/well, adjuvanted Zika MP vaccine (50 µg/well + 25 µg/well, alum MP (50 µg/well), and MPL-A MP (50 µg/well) for 24 h. After the 24 h exposure period, the cells were washed three times to remove extracellular particles. The 96 well plate was then incubated with the MTT reagent (5 mg/mL in PBS) for 2 h. Dimethyl sulfoxide (DMSO) was added after the 2 h in order to dissolve the formazan precipitate. Afterwards, the plate was kept for shaking at room temperature, protected from light, for 20 min. The absorbance was measured at 570 nm using a microplate reader BioTek^®^ Synergy H1 (BIO-TEK Instruments, Winooski, VT, USA). The percent cell viability was calculated relative to the viability of cells treated with growth media only.

### 2.9. Animal Studies

All animal experiments were approved by Mercer University IUCAC protocol (#A2303001). The Zika vaccine microparticles and vaccine MP with adjuvants were administered to 4–6-week-old Swiss Webster mice (Charles River Laboratories, Wilmington, MA). The mice, after a 1 week quarantine period, were randomly assigned as shown in Figure 4A. All experimental mice were given a total of three vaccine doses: a prime dose and two booster doses two weeks apart (week 2 and week 4) via the intranasal (IN) route. The timeline of the in vivo study is shown in Figure 4B. Mice were administered the Zika vaccine via the IN route, and the dose/mouse was 20 µg, MPL-A^®^ was 5 µg, and Alhydrogel^®^ was 20 µg. All mice were challenged with 200 µL of the live Zika virus (Strain PRVABC59, 2.35 × 10^5^ PFU/mL) via intraperitoneal injection 7 weeks after the last booster dose. After challenging the mice, all animals were monitored for 1 week for any weight changes and any physical observations. After 1 week, all mice were sacrificed. Immune organs, such as the spleen and the inguinal and axillary lymph nodes, were extracted for further analysis for cellular response. Serum samples were collected from all mice biweekly. The serum samples were analyzed using enzyme-linked immunosorbent assay (ELISA) for Zika-specific antibodies: IgM, IgG, IgA, IgG2a, IgG1, and IgG3.

### 2.10. Measurement of Antibody Titers Using ELISA

During the animal study, serum was collected from all mice biweekly. As previously described, serum samples were analyzed using enzyme-linked immunosorbent assay (ELISA) for Zika-specific antibodies: IgM, IgG, IgA, IgG2a, IgG1, and IgG3. First, the high-binding polystyrene ELISA plates (Microlon^®^, Greiner Bio-One, Monroe, NC, USA) were coated with the inactivated Zika strain PRVABC59 (50 ng/well) and left at 4 °C overnight. Next, the plate was blocked with 3% BSA (37 °C, 2 h) to prevent nonspecific binding. Next, the serum samples were added to the ELISA plate (37 °C, 2 h) after dilution with PBS. After adding the serum samples, the plate was incubated for 1 hr. After incubation period, HRP-conjugated goat anti-mouse secondary antibody (IgM, or IgG, or IgA, or IgG2a, or IgG1, or IgG3) was added (37 °C, 2 h). The substrate used was 3,3′,5,5′-tetramethylbenzidine (TMB) (BioLegend^®^, San Diego, CA, USA). Sulfuric acid (0.3 M) was used to stop the reaction. Between each step, the plates were washed three times with 0.1% TWEEN 20 in PBS as a wash buffer. After stopping the reaction, the absorbance of the plate was measured using a BioTek^®^ Synergy H1 microplate reader (BIO-TEK Instruments, Winooski, VT, USA) at 450 nm.

### 2.11. Measurement of Cellular, Memory Responses and Intracellular Cytokines

In vivo dosing regimen consisted of a prime dose followed by two booster doses two weeks apart. After dosing, all animals were challenged with live Zika virus via intraperitoneal (IP) injection. After the mice were sacrificed 7 days later, vital immune organs were extracted, such as the spleen, inguinal, and axillary lymph nodes, to determine the T cell and memory responses. All organs collected were processed into single-cell suspensions. All samples were processed through a 40 µm cell strainer. In order to eliminate the red blood cells (RBCs) in the spleen samples, each spleen sample was treated with an ammonium chloride potassium (ACK) lysis buffer. Next, the cells were suspended in Dulbecco’s Modified Eagle Medium (DMEM) supplemented with 70% fetal bovine serum (FBS) and centrifuged at 1500 rpm (20 °C, 10 min). All samples were stored at −80 °C with 5% *v*/*v* dimethyl sulfoxide (DMSO) as a cryoprotectant. To assess the expression of certain markers in lymphocytes and splenocytes, we analyzed the samples using a BD Accuri C6 Plus flow cytometer (BD Bioscience, San Jose, CA, USA). In all samples, the cells were thawed on ice, washed three times, and resuspended in phosphate buffer saline (PBS). After thawing the samples, all cell suspensions were stimulated with 5 µg/mL IL-2 and kept in an incubator overnight. Next day, cell suspensions were treated with 100 µL of anti-mouse APC-labeled CD4 and FITC-labeled CD8a markers in PBS and incubated for 1 h on ice, protected from light. After the incubation period, the cells were washed thrice to remove excess markers. We were interested in assessing the expression of CD4 and CD8a in lymphocytes and splenocytes. We assessed CD45R and CD27 for memory markers and TNF-α, and IL-6 for cytokine markers. When analyzing samples using flow cytometry, the unstained live cells population was gated. A total of 5000 events were accounted for in each sample.

### 2.12. Statistical Analysis

All experiments were performed in triplicates unless otherwise stated. To test for normality, Shapiro–Wilk test was used. To test for variances, a Brown–Forsythe test was employed. For data that followed a normal distribution, a one-way analysis of variance (ANOVA) with Tukey’s post-hoc test was used. For data that did not follow a normal distribution, a nonparametric Kruskal–Wallis test followed by a post-hoc analytical test was conducted. In all cases, the *p* values of <0.05 were considered statistically significant. All the statistical analyses were carried out using GraphPad Prism version 9.2.0 for Windows (GraphPad Software, San Diego, CA, USA, https://www.GraphPad.com). The data are reported as mean ± SEM unless otherwise stated.

## 3. Results

### 3.1. Microparticle (MP) Characterization: ZIKV Vaccine and Adjuvants

The Zika vaccine microparticles were formulated using a double emulsion method as previously described by our lab [1,18]. Zika vaccine MP, Alhydrogel^®^ (alum) MP, and MPL-A^®^ MP were characterized for percent of recovery yield, size of the particles (nm), polydispersity index, zeta potential, and number of particles/mL determined by the laser particle counter (Table 1) [18]. The recovery yield was greater than 85% for Zika MP, Alum MP, and MPL-A MP. The particle size ranged from 400 to 1000 nm. The low PDI indicated a uniform size distribution of the particles. The negative surface charge of Zika vaccine MP, alum MP, and MPL-A MPs indicated no aggregation of MPs.

### 3.2. Fourier Transform Infrared Microscopy (FTIR)

Fourier Transform Infrared Microscopy (FTIR) was utilized for confirming encapsulation of the inactivated Zika antigen in the polymer poly (D, L-lactide-co-glycolide) grade 75:25 (PLGA) [22]. Figure 5 shows the FTIR spectra of Zika solution, Zika microparticles, and PLGA polymer.

### 3.3. Evaluation of In Vitro Immunogenicity Measured via Griess Assay

A Griess assay was utilized to measure immunogenicity of Zika vaccine microparticles in vitro (Figure 6) [1]. We evaluated the Zika MP vaccine, Zika solution, adjuvanted Zika MP vaccine, alum MP, and MPL-A MP by measuring the nitric oxide that was released by dendritic cells 2.4 (DC). Cells only and lipopolysaccharide (LPS) were used as negative and positive controls. There was a significant higher response of Zika MP vaccine than the no treatment group. The adjuvanted Zika MP vaccine was found significantly higher than the unadjuvanted Zika MP vaccine.

### 3.4. Evaluation of In Vitro Cytotoxicity

We measured the cytotoxicity profile of the vaccine microparticles using the MTT assay [1,18]. Dendritic Cells 2.4 (DC) were treated with different treatment groups. Cells only and dimethyl sulfoxide (DMSO) groups served as negative and positive controls. DC cells were treated with Zika solution, Zika MP vaccine, adjuvanted Zika MP vaccine, alum MP, and MPL-A MP for 24 h. The treatment groups were not cytotoxic to the DC cells. Formulated vaccine microparticles were found to be not cytotoxic (Figure 7).

### 3.5. Intranasal Zika-Specific MP Vaccine Humoral Responses

The humoral response was evaluated by using Swiss Webster mice after intranasal administration. The mice were given a prime dose and two booster dose two weeks apart (week 2 and week 4). Serum samples were collected every two weeks. Humoral response was measured via the Enzyme-Linked ImmunoSorbent Assay (ELISA). We measured several antibodies, such as IgM, IgA, IgG, and IgG subtypes (IgG1, IgG2a, and IgG3). IgM is the first antibody that is produced during an infection. Figure 8 displays the IgM antibodies from week 0 to week 11. The IgM antibodies were significant at week 2 and at week 4 where the vaccine groups produced more antibodies than the no treatment group. However, after week 4, the IgM antibodies displayed a decreasing trend. IgG antibodies are the most abundant in serum and crucial for long-term immunity against an infection. The seroconversion from IgM to IgG antibodies is crucial for long-term immunity. Figure 9 shows the Zika-specific IgG titers when measured in Swiss Webster mice. There was a robust total of IgG antibodies found in both ZIKV vaccine and unadjuvanted ZIKV vaccine group. IgG titers were significantly higher in weeks 2, 4, 6, 8, and 11 than the naïve or no treatment group. IgG titers in the adjuvanted and unadjuvanted vaccinated groups were the highest in week 6. Adjuvanted Zika MP vaccine produced a robust IgG antibodies in weeks 2, 4, 6, 8, and 11 than the mice that received no vaccine.

One of the objectives of formulating an intranasal Zika MP vaccine was to determine the presence of IgA antibodies. Possessing mucosal IgA antibodies induced via the intranasal route can be crucial for neutralizing the virus at a potential site of entry for the virus [10]. There were robust IgA titers found after vaccination (Figure 10). Our results show that there was significant production of IgA antibodies in the long-term. Assessing IgA antibodies, the adjuvanted IN Zika MP vaccine induced robust IgA antibodies at weeks 2 and 4, decreasing at weeks 6 and 8, but increasing post-challenge at week 11. The adjuvanted Zika vaccine and unadjuvanted Zika vaccine induced significantly IgA titers than mice that did not receive the vaccine.

To further elucidate the efficiency of the humoral response, we measured the IgG subtypes: IgG1 (Figure 11), IgG2a (Figure 12), and IgG3 (Figure 13). To determine if our Zika MP vaccine induced a Th1- and Th2-mediated immune response, we measured the IgG2a and IgG1 antibodies. A robust IgG1 antibody response is a strong indicator of a significant Th2 response. The IgG1 antibodies (Figure 11) were significantly higher at weeks 2, 4, 6, 8, and peaked at week 11, indicating a robust Th-2 response. The IgG1 antibodies were highest in week 11. The adjuvanted IN vaccine was significantly higher than no treatment group in all weeks. A robust IgG2a antibody titer is a strong indication of a Th-1 response. The IgG2a antibodies were significantly higher than the mice that received no treatment in week 4 and at week 11. The Adjuvated and unadjuvanted vaccinated groups were significantly higher from weeks 2 to 11. After the challenge (week 11), the IgG2a antibodies were significantly higher than the naïve group. These robust IgG2a antibodies establishes that our vaccine MPs can induce a Th1 response. IgG3 antibodies play a vital role in infectious diseases ranging from enhanced control against viruses, increases complement activation, and boosted antibody-dependent cellular cytotoxicity (ADCC) responses [26]. IgG3 antibodies peaked at week 4, then decreased from weeks 6 to 11 (Figure 13). Vaccinated mice showed significantly higher IgG3 subtype titers than the mice that received no treatment.

### 3.6. Intranasal Zika-Specific MP Vaccine Cellular Responses in Spleen and Lymph Nodes

Swiss Webster mice were vaccinated at week 0 as the prime dose followed by two booster doses at weeks 2 and 4. Mice were challenged in week 10. Then, at week 11, the mice were sacrificed and their immune organs, such as the spleen and lymph nodes, were collected to determine the effectiveness of a cellular response. The expression of the CD4^+^ helper and CD8^+^ cytotoxic T cells of the spleen and lymph nodes was analyzed via flow cytometry. In the lymph nodes, our vaccine induced a higher expression of CD4^+^ helper T cells than the CD8^+^ cytotoxic T cells. In Figure 14, the T cell surface markers for CD4^+^ helper T cells (A) and CD8^+^ cytotoxic T cells (B) in lymph nodes are shown. In the lymph nodes, there were strong CD4^+^ and CD8^+^ expressions of T cell markers. The percentage cell counts of CD4^+^ expression (A) were significantly higher than those of the CD8^+^ expression (B). Adjuvanted Zika MP vaccine and Zika MP vaccine percent cell counts were significantly higher than in the no treatment group. In the spleen, the ZIKV vaccine induced a higher expression of CD4^+^ helper T cells than the CD8^+^ cytotoxic T cells. In Figure 15, the expression of CD4^+^ helper T cells (A) and expression of CD8^+^ cytotoxic T cells (B) in the spleen is displayed. In the spleen, the adjuvanted Zika vaccine induced significantly higher CD4^+^ helper T cells and CD8^+^ cytotoxic T cells than in the mice that did not receive the vaccine.

### 3.7. Zika-Specific Memory Response Induced by ZIKV MP Vaccine

To evaluate the memory response of our Zika MP vaccine, we investigated two memory B cells markers, CD45R and CD27, in the spleen and lymph nodes (Figure 16). There was robust expression of memory response in the spleen and lymph nodes. In the lymph nodes, there was a significant memory response seen in both CD27 and CD45R (Figure 16A,B). IN adjuvanted Zika MP vaccine and the unadjuvanted Zika vaccine were both significantly higher than in the no treatment group for both the memory makers. In the spleen, there was a significant memory response seen in both CD27 and CD45R (Figure 16C,D). The adjuvanted Zika MP vaccine and the unadjuvanted demonstrated higher memory response than the no treatment group. The expression of CD27 was significantly higher in both the spleen and the lymph nodes (A and C). The expression of CD45R was significantly greater in the spleen and the lymph nodes (B and D). Regarding CD45R expression in the lymph nodes (B), there was no significant difference between the adjuvanted Zika MP vaccine versus the Zika MP vaccine. The expression of CD45R in the adjuvanted Zika MP vaccine and the Zika MP vaccine was significantly higher (D) than in the no treatment group.

### 3.8. Zika-Specific Intracellular Cytokines Produced via ZIKV MP Vaccine

In addition, we tested the efficiency of our Zika MP vaccine to induce intracellular cytokines, Interleukin-6 (IL-6) and Tumor Necrosis Factor alpha (TNF-α) in the spleen. The vaccine produced a robust response of IL-6 and TNF- α in the spleen (Figure 17A,B). Interestingly, our Zika microparticulate vaccine induced a significantly higher expression or percentage of cell count of IL-6 (A) than TNF- α (B). The adjuvanted Zika MP vaccine induced a significantly higher expression of intracellular cytokines, IL-6 and TNF- α, than in the no treatment group (A and B) when the vaccine was administered intranasally.

## 4. Discussion

There are several complications of this single stranded RNA virus, which renders this as an important virus to investigate for researchers and clinicians. In infants, the major clinical observation was microencephaly. However, in adults, the infection from Zika virus has been associated with Guillain–Barré syndrome (GBS). GBS can affect individuals at any age, but most commonly affects adults who are between 30 and 50 years old. To date, there are no approved treatments or vaccines available. In this proof-of-concept study, we explored a pain-free microparticulate vaccine that can be administered intranasally.

A needle-free approach can offer several benefits such as no pain, reduction in anxiety, decreases vaccine hesitancy, and can improve patient compliance [27,28]. Traditional intramuscular (IM) vaccines have several conventional side effects such as pain at the injection site, swelling, redness, fatigue, headache, chills, fever, and motion sickness. Along with IM vaccines, some individuals have a fear of needles. In a recent study investigating needle fear or anxiety, children, approximately 20–30% of young adults, and 20–50% of adolescents displayed a fear of needles [29]. An intranasal vaccine can curb the anxiety revolving around needles among many patient populations. This approach would decrease refusal or hesitancy and encourage more patient compliance for vaccinations.

The route of administration is a crucial element when formulating a vaccine. Understanding clinical and laboratory findings can aid in narrowing potential routes of interest. In recent findings, ZIKV has been discovered in saliva, urine, body fluids, breast milk, semen, and in nasopharyngeal swabs [9,10]. Sustained secretion of Zika in lacrimal fluids and in nasal mucosa suggests that close contact with an individual or exposure to the infectious body fluids could transmit and cause infection in others. A recent study investigating guinea pigs found that ZIKV is transmitted via close contact as evident by the detection of viremia and the viral secretion in tears and saliva [10]. Similar findings were found when investigating the intranasal infection spread of ZIKV in macaques [10]. The intranasal (IN) route of vaccine delivery has gained increased popularity over the years. Vaccine delivery via the nasal cavity [24] is beneficial since the mucosa has an abundance of lymphatic tissue or nasal-associated lymphoid tissue (NALT) and specialized cells called microfold (M) cells (Figure 2). This route is attractive because it induces a systemic and local mucosal immune response [27].

In our approach, we formulated and investigated a microparticulate Zika vaccine (Figure 3). A microparticulate or particulate vaccine formulation has several benefits. Generally, particulate formulations are more immunogenic than soluble antigens due to the less effective cross-presentation [18,20,30,31]. In addition, the majority of soluble antigens are not recognized, and less endocytosis occurs by the APCs rendering less protective immunity against the pathogen. However, the encapsulation or conjugation of soluble antigens with biodegradable carriers can improve their recognition and the uptake process by APCs [17,32]. Particulate carriers, such as PLGA, serve as effective antigen delivery systems that enhance and/or facilitate the uptake of antigens by various antigen-presenting cells (APCs) like dendritic cells (DCs) or macrophages. The encapsulation of the antigen or inactivated Zika virus can protect the structure against proteolytic degradation and facilitate the antigen delivery to the APCs. In our previous paper, we showed that the process for encapsulation did not affect the Zika structure [1]. The microparticles provided a sustained release of the antigen [1] leading to a robust innate and adaptive immune response [21,33]. Stability has another limitation for consideration when formulating a vaccine [28,34,35]. However, a microparticulate formulation overcomes this limitation by avoiding the cold-chain storage issue. IM vaccines that are in the market are usually in liquid form, and preservation of the vials requires equipment that can provide lower temperatures to conserve the shelf-life of the vaccines. In our microparticulate approach, the inactivated Zika virus was used as the antigen. In order to increase the immunogenicity response, FDA-approved adjuvants were used in our study. Alhydrogel^®^ has been shown to induce a Th2-mediated immune response, while MPL-A^®^ induces a Th1-mediated response. In this approach, we explored an intranasal Zika microparticulate vaccine along with adjuvants providing a balanced Th1 with Th2 immune response.

The microparticulate Zika vaccine was formulated via a double emulsion method developed by our lab [18,21,36]. The emulsion was lyophilized, then the microparticles were characterized for recovery yield, particle size (nm), polydispersity index (PDI), number of particles/mL determined by the laser particle counter [18], and zeta potential (mV). We observed a high recovery yield (Table 1) that was above 85% for the Zika MP vaccine, Alhydrogel (alum) MPs, and MPL-A MPs. The particle size of all vaccine microparticles ranged from 400 to 1000 nm, which is in the range for uptake by APCs to process the microparticles and stimulate an innate immune response. The PDI was low, which suggests the particles had a uniform size distribution. The charge or zeta potential of the Zika MP vaccine and adjuvants (alum and MPL-A) was negative, which demonstrates no aggregation of the microparticles. The Zika MP vaccine had an average of 1180 particles per mL. To ensure the successful encapsulation of the inactivated Zika vaccine into the PLGA polymer, we employed the FITR technique. In Figure 5, the FTIR spectra of Zika solution (black color), PLGA (red color), Zika microparticles (blue color) is shown. In the Zika solution, one major peak of the FTIR spectra (black) can be observed at 1648 cm^−1^. These observations are consistent with the reported literature, with peaks at 1550 cm^−1^ and 1630 cm^−1^ [23,37]. The poly (D, L-lactide-co-glycolide) grade 75:25 (PLGA) polymer (red) [24] showed, as expected, small but visible peaks at 2954 cm^−1^, 2833 cm^−1^, 1758 cm^−1^, and 1093 cm^−1^. Interestingly, the Zika MPs spectra (blue) displayed peaks similar to the PLGA polymer at 3348 cm^−1^ (OH bond), 2930 cm^−1^ (CH), 2855 cm^−1^, 1745 cm^−1^ (carbonyl bond), 1085 cm^−1^ (CO bond), and 1550 cm^−1^. The peak representing the Zika solution at 1648 cm^−1^ was notably decreased. In a recent study, researchers were interested in identifying an infrared profile to detect ZIKV signatures that are present in saliva that can be used for screening or for diagnostic purposes [23]. Since saliva can contain proteins, for instance mRNA, it can also contain the presence of ZIKV. They found that the 1547 cm^−1^ peak in FTIR indicates that amide II can aid in serving as a biomarker for detecting if an individual has a ZIKV infection. Our results show a similar peak at 1550 cm^−1^, which is close to the reported peak, suggesting this peak indeed may serve as a identifying marker for Zika. FTIR spectra results suggest that the inactivated Zika antigen was successfully encapsulated within the biodegradable PLGA polymer.

In vitro immunostimulatory potential was conducted using the Griess assay [18,30,36]. The release of nitric oxide (NO) was analyzed by quantitating the oxidation product, nitrite, via this assay [18,36]. In this test, the release of NO by dendritic cells (DC 2.4) helps to determine if the vaccine microparticles are immunogenic. Further, once the Zika microparticles are identified and up taken by the antigen-presenting, they release a non-specific innate immune marker NO along with other cytokines, such as IL-12, TNF, and IFN-γ. We evaluated the Zika MP vaccine, Zika solution, adjuvanted Zika MP vaccine, alum MP, and MPL-A MP (Figure 6). There was a significant response of Zika MP vaccine than the no treatment group. The adjuvanted Zika MP vaccine (alum MP and MPL-A MP) was found significantly higher than the unadjuvanted Zika MP vaccine. However, there was no significant difference between the Zika solution and the no treatment group. Taken together, the adjuvanted Zika MP vaccine produced the highest innate response. In addition to measuring the innate response, the cytotoxicity profile of the Zika vaccine is vital. The MTT assay was utilized to determine the cytotoxicity profile of the Zika vaccine microparticles. This assay assesses the conversion of MTT into formazan crystals by viable or living cells [18,38]. Dendritic Cells 2.4 (DC) were treated with various treatment groups. DC cells were treated with Zika solution, Zika MP vaccine, adjuvanted Zika MP vaccine, alum MP, and MPL-A MP for 24 h (Figure 7). Zika in solution form and vaccine microparticles (Zika and adjuvants) were not cytotoxic to the DC cells. Cytotoxicity results suggest that the formulated vaccine microparticles are not cytotoxic.

The intranasal Zika microparticulate vaccine demonstrated a strong production of humoral antibodies. During an infection, the first antibody that is produced is IgM. There were robust IgM antibodies (Figure 8) produced during the initial weeks (2 and 4). There were significantly higher IgM antibodies in the IN adjuvanted Zika MP vaccine than the no treatment group. As expected, the production of IgM antibodies significantly decreased after week 4, but was complemented with total IgG titers. The long-term IgG antibodies displayed an increasing trend from week 2 to week 11 (Figure 9). The adjuvanted IN Zika MP vaccine induced robust and extended IgG antibodies in weeks 2–11 compared to the mice group that received no vaccine. These results showed a strong seroconversion from IgM to IgG antibodies with the particulate vaccine administered intranasally. We observed similar results when we previously investigated a microparticulate measles vaccine demonstrating a strong production of total IgG titers [31]. An important hallmark of an intranasal vaccination strategy is the production of Zika-specific IgA antibodies. IgA antibodies can protect the mucosal surfaces to limit the spread of the infection. The nasal mucosa is constantly exposed to numerous antigens, and the NALT (nasal-associated lymphoid tissue) is generally considered a vital site for inducing mucosal immunity due to the production of IgA within the nasal cavity [39]. Interestingly, nasal vaccines when administered in humans have been demonstrated to induce significant IgA and IgG titers in other mucosal areas, like the vagina with sera titers that were similar to when individuals were administered a direct intravaginal vaccine [39]. Our vaccine demonstrated the long-term production of excellent IgA titers. Assessing IgA antibodies (Figure 10), the adjuvanted IN Zika MP vaccine induced robust IgA antibodies at week 4, decreasing at weeks 6 and 8, but increasing post-challenge at week 11. We further explored the humoral response by assessing the presence of the IgG subtypes: IgG1 and IgG2a. The IgG1 subtype has a unique array of functions, such as eliciting an antibody response against viral pathogens, having the highest FcγR-binding affinity, and participating in antibody-dependent cellular cytotoxicity (ADCC) [40,41]. IgG2a antibodies can trigger responses against polysaccharides and DNA or RNA viruses. Measuring IgG1 and IgG2a Zika-specific antibodies allows us to determine if our microparticulate Zika vaccine produced a balanced Th2 and Th1 cell-mediated immune response. A balanced Th1 and Th2 immune response is vital for a healthy immune system. Recent literature suggests that an imbalance can have detrimental outcomes. Researchers have shown that an elevated Th1/Th2 ratio increases the cytotoxicity against an embryo and can lead to implantation failure. A proper ratio balance may be needed for better reproductive outcomes [42]. Another reported consequence of an imbalance in the ratio is related to an immune dysregulation association with an HIV infection [43]. Th1/Th2 ratio imbalance caused natural killer (NK) cell activity, differentiation, and cytotoxic CD8^+^ T cell activation to decrease significantly. Our results suggest that there was a balanced Th1- and Th2-mediated response. We found a high IgG1 (Figure 11), IgG2a (Figure 12) antibodies in all vaccinated groups than the no treatment group. The adjuvanted Zika MP vaccine antibody titers were significantly higher than the unadjuvanted and no treatment group suggesting that an adjuvant may be needed to format an IN vaccine. Another vital subclass of IgG is IgG3, a unique potent immunoglobulin, that can protect against a vast range of pathogens from bacteria to viruses. IgG3 antibodies has several functions like potent mediators of effector functions, triggers complement activation, enhanced neutralization, and improved antibody-mediated cellular cytotoxicity (ADCC) [26]. Our vaccine results good IgG3 titers (Figure 13) for both vaccinated groups. According to the reported literature, IgG3 was shown to increase in weeks 2–4, and then the titers decreased. Ig3 antibody expression, in our data, shows IgG3 peak at week 4, then gradually increases. This is consistent with the other reported literature [44]. As compared to other IgG or its subtypes, IgG3 has a 7-day half-life mainly due to its relative lower affinity to the neonate Fc receptor (FcRn). As a result, IgG3 degradation gradually increases, and thus, slows down its effector functions. This limits the detection of IgG3 in peripheral blood from the mice. However, Zika-specific IgG3 expression has been shown as a reliable biomarker of recent viral exposure [44]. Our results show a good humoral response, especially in the adjuvanted group, suggesting that adding an adjuvant to regimen may be needed. In our vaccine, we used two FDA-approved adjuvants (alum and MPL-A) in microparticulate form. Alum works by depot effect, promotes pro-phagocytic effects, and increases immuno-stimulation via the NLRP3 pathway [20]. MPL-A is a toll-like-receptor (TLR) 4 that is involved in promoting phagocytosis by APCs. In several phase I studies, the ZIKV vaccine, when administered to adults, showed safety and tolerability, but immunogenicity diminished for long-term use. In a phase I study, researchers used the mRNA platform to develop the Zika vaccine without any adjuvant and tested the two-dose method with three different doses (10, 25, and 100 μg) of the vaccine administered to patients [45]. Although all three doses were well tolerated by patients, this vaccine approach produced inadequate Zika virus-specific neutralizing antibody responses. They found geometric mean titers (GMTs) were greatest, but only after dose two. There have been others reports of using a booster dose to enhance the immune responses. Another phase I study used purified inactivated Zika virus vaccine given intramuscularly [46]. As expected, pain at the injection site was noted as an adverse event. After a prime dose, humoral responses and seroconversion was very low. However, a third Zika dose overcame this limitation where the humoral responses climbed significantly after the third dose with a 100% rate of seroconversion. In another phase I study, researchers investigated an adjuvanted, inactivated, purified whole-virus Zika vaccine with two different immunization regimens (high or low dose). They found similar findings with other clinical trials where they found higher doses of the Zika vaccine and a booster dose may be needed to optimize antibody persistence. In our proof-of-concept study, the Zika MP vaccine and the adjuvated Zika vaccine, with the prime and two booster dose method when administered intranasally, produced significant immune humoral responses.

We investigated the cellular response of the adjuvanted Zika MP and Zika MP vaccine by analyzing CD4^+^ helper and CD8^+^ cytotoxic T cells. We investigated the lymph nodes and the spleen to determine the effectiveness of the Zika vaccine. We found excellent T-cell surface marker expression in both immune organs for both the adjuvanted Zika MP and Zika MP vaccine. In the lymph nodes (LN), there was a significantly high expression or percentage of CD4^+^ helper T cells (Figure 14A) and CD8^+^ cytotoxic T cells (Figure 14B). In the LNs, the expression of helper CD4^+^ and cytotoxic CD8^+^ T cells was significantly higher in the adjuvanted Zika MP vaccine than in the no treatment group, suggesting that the booster strategy was effective. An increased level of CD4^+^ helper T cells is beneficial due to the various functions of the helper cells. CD4^+^ helper T cells can differentiate into T helper 1 (Th1) effectors to induce CD8^+^ T cells and into T follicular helper (Tfh) cells to promote the generation of antibodies, which can neutralize the Zika virus [47]. In addition, CD4^+^ helper T cells can create several cytokines to increase the communication process between innate cells, such as macrophages, or natural killer (NK) cells. NK cells are essential for identifying and lysing the Zika-infected cells. After binding to the stressed or Zika-infected cell, the NK cell releases the cytotoxic granules directly into the target cell to kill the cell [48]. There was a high expression of CD4^+^ helper T cells (Figure 15A) and CD8^+^ cytotoxic T cells (Figure 15B) in the spleen. The adjuvanted Zika MP vaccine displayed a higher expression of helper CD4^+^ and cytotoxic CD8^+^ T cells than the unadjuvanted Zika MP vaccine or naïve group indicating that the adjuvanted Zika MP vaccine is effective in producing a cellular immune response.

To further evaluate the effectiveness of our Zika MP vaccine, we measured several markers for memory response and intracellular cytokines. In our study, we investigated two memory B cells markers, CD45R and CD27, in spleen and lymph nodes (Figure 16). The expression of CD45R was significantly higher in both the spleen (Figure 16D) and lymph nodes (Figure 16B). The expression of CD27 memory response was significantly higher in the spleen (Figure 16C) than the lymph nodes (Figure 16A). The adjuvanted Zika MP vaccine triggered a greater memory response than the unadjuvanted Zika MP vaccine. Intracellular cytokines play an essential role in stimulating and prolonging the immune response. In the reported literature, researchers investigated the correlation of their intranasal vaccine, recombinant ZIKV envelope DIII (ZDIII) protein that was genetically fused with *Salmonella typhimurium* flagellin (FliC-ZDIII), and the number of doses required to generate significant immune responses [49]. After the third dose, the FliC-ZDIII vaccine produced significant IFN-γ, IL-4, IL17A, and IL-22 cellular responses in the spleen suggesting that the one dose strategy may not be adequate for generating a robust immune response [49]. Our results show the Zika MP vaccine and adjuvanted Zika MP vaccine produced significant expression or percentage of cells of Interleukin-6 (IL-6) (Figure 17A) and Tumor Necrosis Factor alpha (TNF-α) (Figure 17B) in the spleen. Mature B cells produce IL-6, and this secretion further triggers immune cells to increase the production of other cytokines [50]. The literature has also noted that IL-6 promotes the survival of T cells by preventing the activation of induced cell death by decreasing the expression of Fas and FasL receptors [51]. Several immune responses such as macrophages, lymphocytes, and endothelial cells produce TNF-α [52,53]. In our investigation, we found a robust response of intracellular cytokines in the spleen. The adjuvanted Zika MP vaccine produced significantly higher memory makers than the unadjuvanted and no treatment group. Therefore, intranasal immunization with adjuvanted Zika microparticulate vaccine and Zika microparticulate vaccine induced potent memory and intracellular cytokines.

Although the mice were challenged with a live virus, the dose was a sub-lethal one. We did not observe any significant weight changes. In our future study, we will challenge with a lethal dose of the live virus and investigate the neutralization capacity of our Zika vaccine. We will investigate other Zika-specific memory markers along with intracellular cytokines to understand the detailed cellular interactions that can help to booster an even stronger cellular and long-term memory response. In this pre-clinical study, the expression of two intracellular cytokines was seen in the spleen. However, there was no significant difference when we tested the same intracellular cytokines in lymph nodes indicating that there was a very low expression of cytokines. There are several other organs which can be further explored to determine how Zika affects the cells in those organs, such as the brain. To understand the intricate complexity of the communication process of various cytokines, we will evaluate the expression of Zika-specific cytokines that are vital for immune cells such as NK cells, macrophages, and different T cells which will help uncover the mechanism how our Zika microparticulate vaccine may work in an infection.

## 5. Conclusions

Since the outbreak of Zika worldwide, there has been tremendous research into the Zika virus. Currently, no vaccine or treatment exists. In our proof-of-concept study, we explored a microparticulate Zika vaccine which can protect the antigen and offer a sustained release to further enhance the stimulation of immune responses. Unlike the traditional needle method, we investigated a pain-free Zika vaccine administered via intranasal immunization. Exploring pain-free alternatives may help increase compliance and vaccination rates worldwide. Our results indicate that the Zika MP vaccine produced a sustained antibody response, but with the help of adjuvants, the cellular and memory response was improved. In addition, as the other literature has demonstrated, more than one dose of the vaccine may be needed to generate long-term immunity and cross-protectivity. In this proof-of-concept study, we established the feasibility of formulating an intranasal Zika microparticulate vaccine that demonstrates strengthening the humoral and cellular responses.

## Figures and Tables

**Figure 1 viruses-16-00865-f001:**
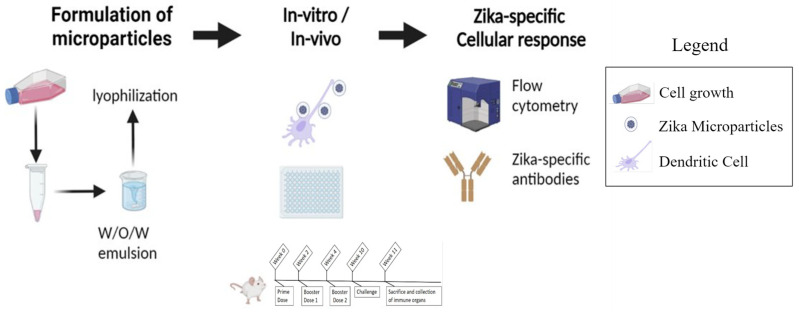
Overview of formulation of microparticles, in vitro, in vivo assessment, and measurement of Zika-specific immune responses.

**Figure 2 viruses-16-00865-f002:**
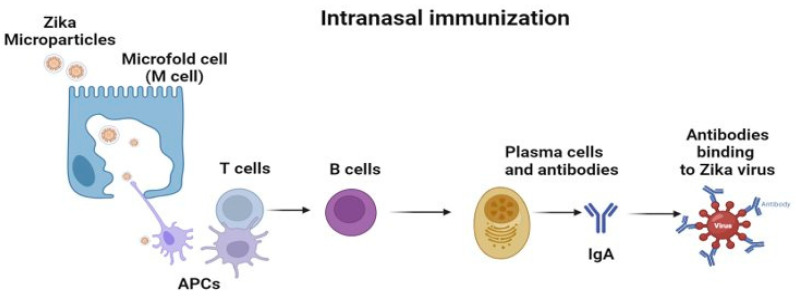
Intranasal immunization with Zika microparticles. M cells can uptake the Zika MPs via dendritic cells (DCs) or APCs. The particles that were up taken, processed, and reflected on the Major Histocompatibility Complex I/II (MHC I/II) are presented to T cells. The T cells can activate B cells to proliferate into plasma cells to produce antibodies such as IgA for mucosal immunity.

**Figure 3 viruses-16-00865-f003:**
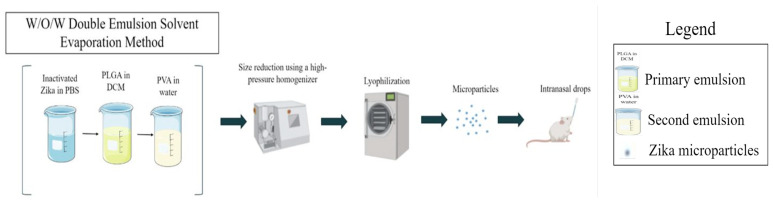
Formulation of Zika microparticles using the w/o/w double emulsion solvent and evaporation method.

**Figure 4 viruses-16-00865-f004:**
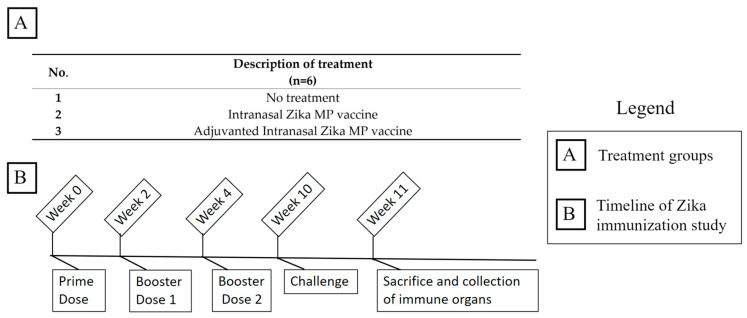
(**A**): Groups of mice (*n* = 6) in the in vivo efficacy evaluation of intranasal Zika MP vaccine. (**B**): Timeline of the in vivo study assessing the Zika vaccine administered via the intranasal route.

**Figure 5 viruses-16-00865-f005:**
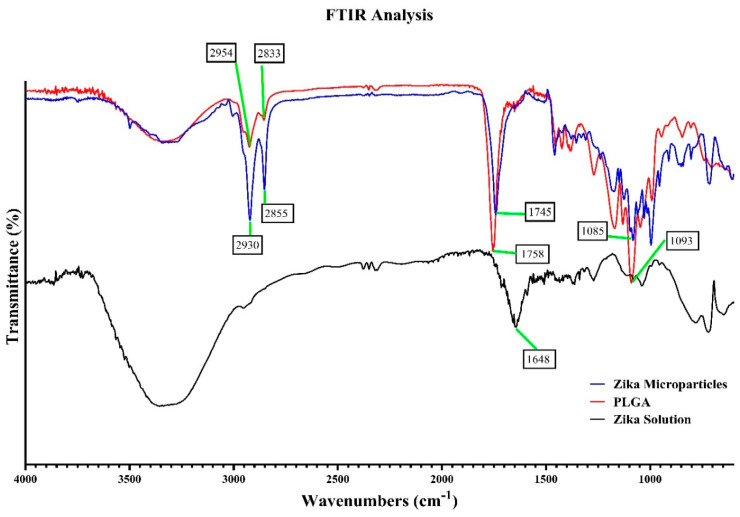
FTIR spectra of Zika solution (black color), PLGA (red color), Zika microparticles (blue color). In the Zika solution FTIR spectra (black), one main peak is seen at 1648 cm^−1^. These observations are consistent with the reported literature [23]. Small, but visible peaks at 2954 cm^−1^, 2833 cm^−1^, 1758 cm^−1^, and 1093 cm^−1^ are representative of the poly (D, L-lactide-co-glycolide) grade 75:25 (PLGA) polymer [24]. In the Zika microparticle spectra (blue), the main peaks observed at 2930 cm^−1^, 2855 cm^−1^, 1745 cm^−1^, 1085 cm^−1^, 1550 cm^−1^). Based on the FTIR spectra, the inactivated Zika antigen was successfully encapsulated within the biodegradable PLGA polymer.

**Figure 6 viruses-16-00865-f006:**
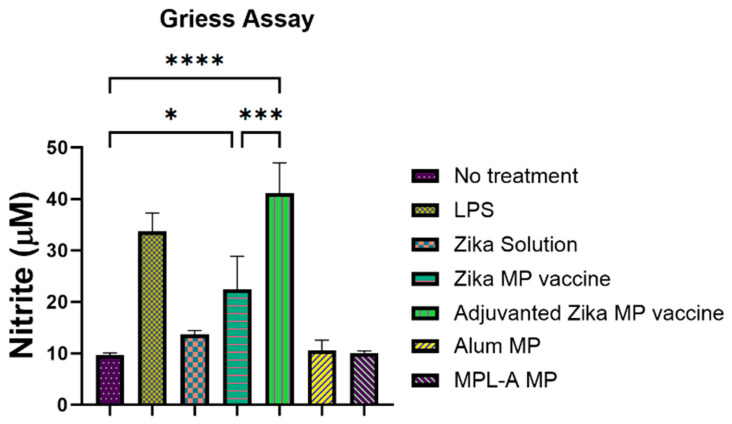
Measurement of nitric oxide released by murine dendritic cells (DC 2.4) by Griess assay [25]. Murine dendritic cells exposed to vaccine MP with and without adjuvants produced significantly higher nitric oxide than the cells that did not receive any treatment. Cells exposed to adjuvant MP (Alum MP and MPL-A MP) did not produce significant amount of nitric oxide. Unstimulated cells or cells were used as negative control. Lipopolysaccharide (LPS) was used as a positive control. Inactivated Zika solution (50 µg/well), Zika MP vaccine (50 µg/well), Zika MP (50 µg/well) + adjuvant (25 µg/well), alum MP (50 µg/well), and MPL-A (50 µg/well). (Data represented as Mean ± SEM, one-way ANOVA test followed by Dunnett’s multiple comparison test; *, *p* < 0.05, ***, *p* < 0.001, ****, *p* < 0.0001).

**Figure 7 viruses-16-00865-f007:**
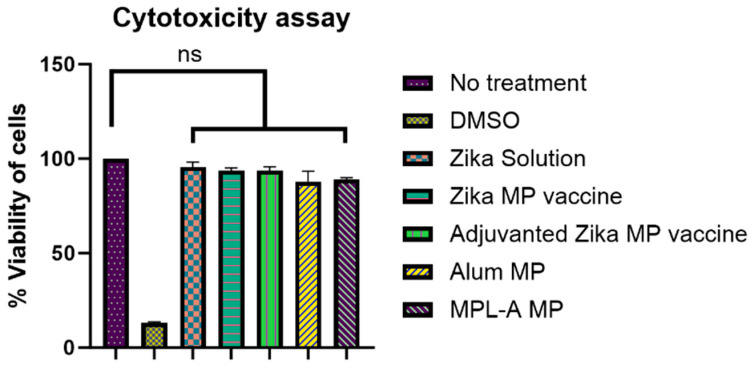
Evaluation of in vitro cytotoxicity of microparticles using MTT assay in murine dendritic cells (DC 2.4). Cells treated with inactivated Zika solution (50 µg/well), Zika MP vaccine (50 µg/well), Zika MP (50 µg/well) + adjuvant (25 µg/well), alum MP (50 µg/well), and MPL-A (50 µg/well) for 24 h. Cells were viable after 24 h of exposure. Cells treated with DMSO showed reduction in viability of cells. Data represented as Mean ± SEM, one-way ANOVA test followed by Dunnett’s multiple comparison test; ns, non-significant.

**Figure 8 viruses-16-00865-f008:**
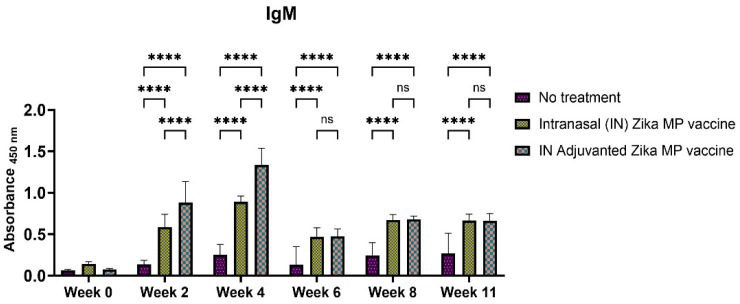
Zika-specific IgM titer measurement in serum using ELISA [25]. Animals received one prime dose at week 0 and two booster doses on weeks 2 and 4 via intranasal administration. Mice that received an intranasal MP vaccine and adjuvanted vaccine MP induced significantly higher antibody titers than the untreated control group (weeks 2, 4). Moreover, mice receiving adjuvanted vaccine MP induced significantly higher antibody titers than the mice receiving the MP vaccine (weeks 2, 4). IgM titers decreased significantly after week 4. Data represented as Mean ± SEM and a Brown–Forsythe ANOVA test, followed by Tukey’s multiple comparison test; ****, *p* < 0.0001. ns, non-significant.

**Figure 9 viruses-16-00865-f009:**
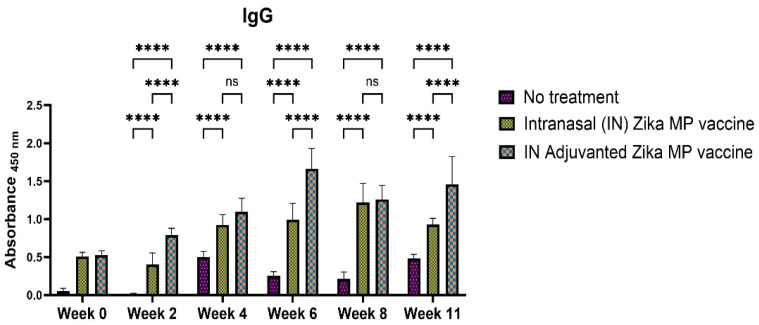
Zika-specific IgG titer measurement in serum using ELISA [25]. Animals received one prime dose at week 0 and two booster doses on weeks 2 and 4 via intranasal administration. Mice that received an intranasal MP vaccine and adjuvanted vaccine MP induced significantly higher antibody titers than the untreated control group (weeks 2, 4, 6, 8, and 11). Adjuvanted Zika vaccine MP induced significantly higher antibody titers than the mice receiving the Zika MP vaccine (weeks 2, 6, and 11). (Data represented as Mean ± SEM and a Brown–Forsythe ANOVA test, followed by Tukey’s multiple comparison test; ****, *p* < 0.0001). ns, non-significant.

**Figure 10 viruses-16-00865-f010:**
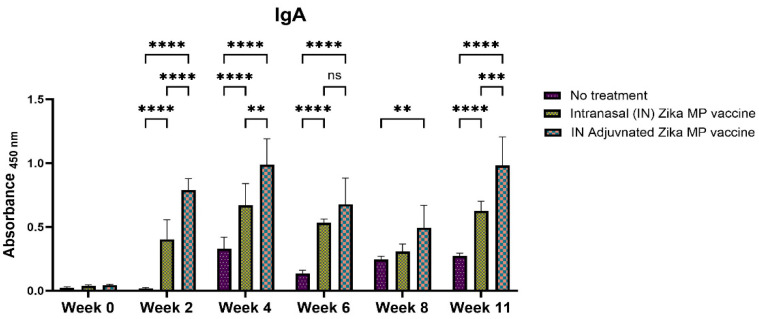
Zika-specific IgA titer measurement in serum using ELISA [25]. Mice received one prime dose at week 0 and two booster doses at weeks 2 and 4 via intranasal administration. Mice that received an intranasal MP vaccine and adjuvanted vaccine MP induced significantly higher antibody titers than the no treatment group (weeks 2–11). Moreover, mice receiving adjuvanted Zika MP vaccine displayed significantly higher antibody titers than the mice receiving the MP vaccine (weeks 2, 4, and 11). (Data represented as Mean ± SEM and a Brown–Forsythe ANOVA test, followed by Tukey’s multiple comparison test; **, *p* < 0.01, ***, *p* < 0.001, ****, *p* < 0.0001). ns, non-significant.

**Figure 11 viruses-16-00865-f011:**
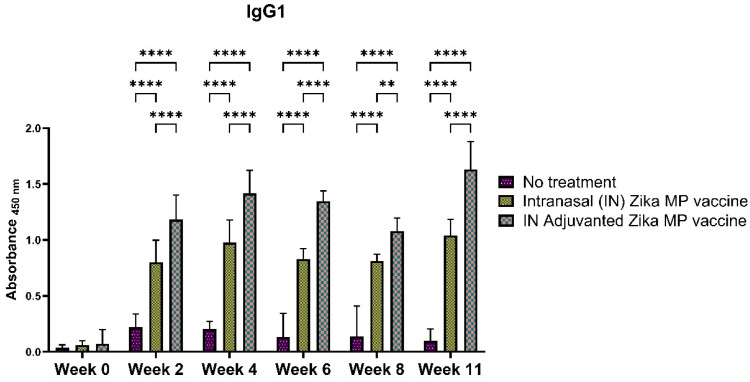
Zika-specific IgG1 titer measurement in serum using ELISA [25]. Mice received one prime dose at week 0 and two booster doses at weeks 2 and 4 via intranasal administration. We found a robust IgG1 antibody response indicating of a significant Th2 response. Mice that received an intranasal MP vaccine and adjuvanted vaccine MP induced significantly higher antibody titers than the no treatment group (weeks 2–11). Moreover, mice that received the adjuvanted Zika MP vaccine showed significantly higher antibody titers than the mice receiving the Zika MP vaccine (weeks 2–11). (Data represented as Mean ± SEM and a Brown–Forsythe ANOVA test, followed by Tukey’s multiple comparison test; **, *p* < 0.01, ****, *p* < 0.0001).

**Figure 12 viruses-16-00865-f012:**
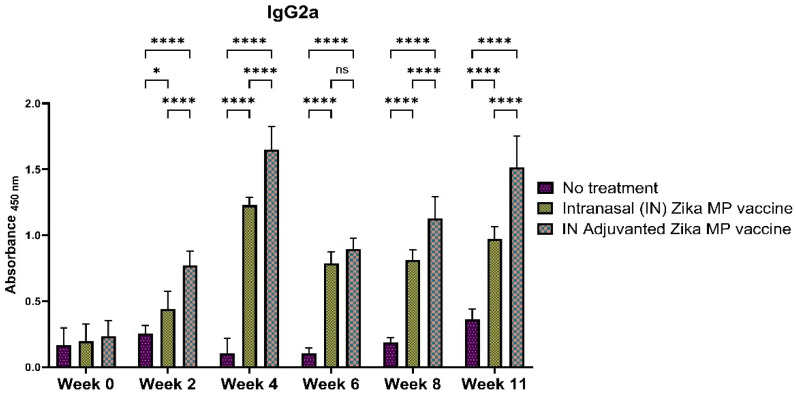
Zika-specific IgG2a titer measurement in serum using ELISA [25]. Mice received one prime dose at week 0 and two booster doses at weeks 2 and 4 via intranasal administration. There was a strong IgG1 antibody titer suggesting a significant Th2 response. Mice that received an intranasal Zika MP vaccine and adjuvanted vaccine MP induced significantly higher antibody titers than the no treatment group (weeks 2–11). Moreover, mice receiving adjuvanted vaccine MP induced significantly higher antibody titers than the mice receiving the Zika MP vaccine (weeks 2, 4, 8, and 11). (Data represented as Mean ± SEM and a Brown–Forsythe ANOVA test, followed by Tukey’s multiple comparison test; ns, non-significant, *, *p* < 0.05, ****, *p* < 0.0001).

**Figure 13 viruses-16-00865-f013:**
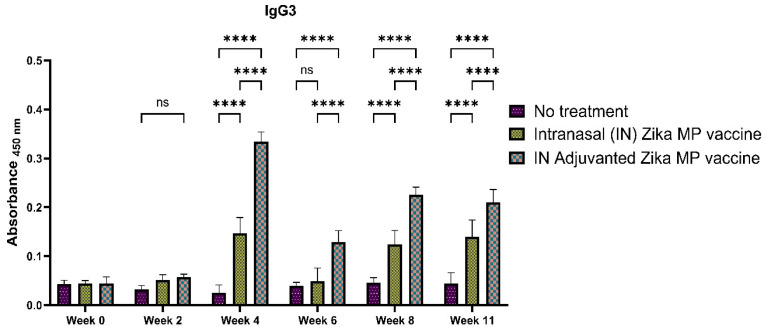
Zika-specific IgG3 titer measured in serum using ELISA [25]. Mice received one prime dose at week 0 and two booster doses at weeks 2 and 4 via intranasal administration. There was a strong Ig3 titers found in the vaccinated mice versus the no treatment group. IgG3 antibody titers peaked at week 4, then decreased in subsequent weeks. Adjuvanted Zika MP vaccine was significantly higher than the no treatment group in weeks 2–11. (Data represented as Mean ± SEM and a Brown–Forsythe ANOVA test, followed by Tukey’s multiple comparison test; ns, non-significant, ****, *p* < 0.0001).

**Figure 14 viruses-16-00865-f014:**
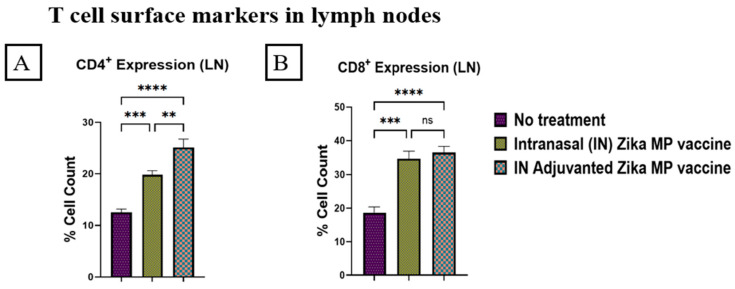
Zika-specific expression of CD4^+^ helper (**A**) and CD8^+^ cytotoxic (**B**) T cells in lymph nodes measured via flow cytometry. Swiss Webster mice were given a prime dose at week 0, then followed by two boosters two weeks apart (weeks 2 and 4). Lymph nodes were collected at week 11. The vaccine produced significant CD4^+^ helper and CD8^+^ cytotoxic T cell surface marker cellular response after vaccination. Expression or percentage of cell count of CD4^+^ helper T cells was significantly higher than the expression of CD8^+^ cytotoxic T cells in lymph nodes. Expression of CD4^+^ and CD8^+^ T cells was significantly higher in the IN adjuvanted Zika MP vaccine group than mice that received no treatment. (Data represented as Mean ± SEM and a Brown–Forsythe ANOVA test, followed by Tukey’s multiple comparison test; ns, non-significant, **, *p* < 0.01, ***, *p* < 0.001, ****, *p* < 0.0001).

**Figure 15 viruses-16-00865-f015:**
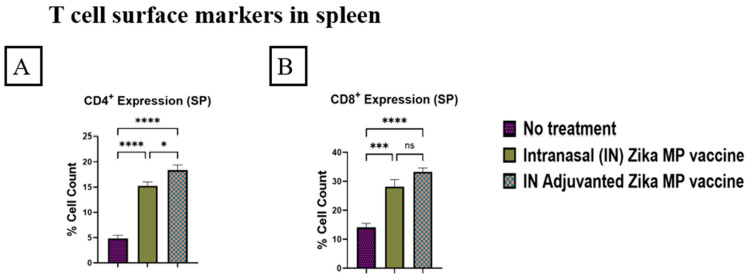
Zika-specific expression of CD4^+^ helper (**A**) and CD8^+^ cytotoxic (**B**) T cells in the spleen measured via flow cytometry. Swiss Webster mice were given a prime dose at week 0, then followed by two boosters two weeks apart (weeks 2 and 4). The spleen was collected at week 11. The vaccine produced significant CD4^+^ helper and CD8^+^ cytotoxic T cell surface marker cellular response after vaccination in the spleen. The percentage of cells of CD4^+^ helper T cells was significantly higher than the expression of CD8^+^ cytotoxic T cells in the spleen. Expression of CD4^+^ and CD8^+^ T cells was significantly higher in the IN adjuvanted Zika MP vaccine group than the group that received no treatment. (Data represented as Mean ± SEM and a Brown–Forsythe ANOVA test, followed by Tukey’s multiple comparison test; ns, non-significant, *, *p* < 0.1, ***, *p* < 0.001, ****, *p* < 0.0001).

**Figure 16 viruses-16-00865-f016:**
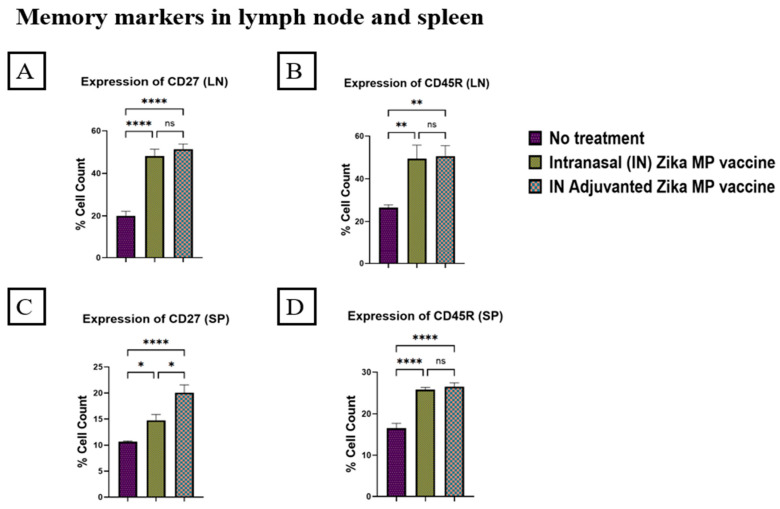
Zika-specific expression of memory markers (CD27 and CD45R) in the lymph nodes and spleen. Graph (**A**): expression or percentage of cell count of CD27 in the lymph nodes, (**B**): expression of CD45R in the lymph nodes, (**C**): expression of CD27 in the spleen, and (**D**): expression of CD45R in the spleen. Mice were administered one prime dose at week 0 and two booster doses at weeks 2 and 4 via intranasal administration. Immune organs, such as the lymph nodes and spleen, were collected at week 11 and analyzed via flow cytometry. The Zika vaccine, adjuvanted and unadjuvanted, demonstrated strong memory response in the spleen and the lymph nodes. The adjuvanted Zika MP vaccine and Zika MP vaccine expressed higher levels of memory markers than in the no treatment group. (Data represented as Mean ± SEM and a Brown–Forsythe ANOVA test, followed by Dunnett’s multiple comparison test; ns, non-significant, *, *p* < 0.1, **, *p* < 0.01, ****, *p* < 0.0001).

**Figure 17 viruses-16-00865-f017:**
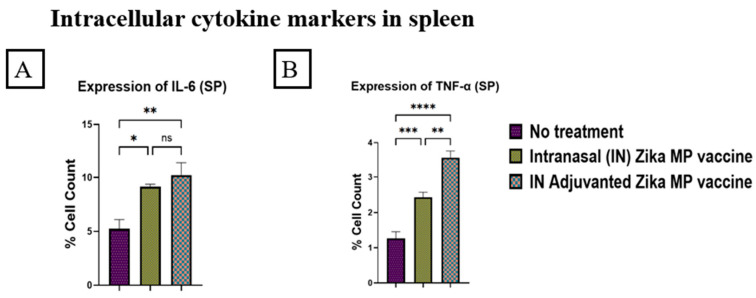
Zika-specific expression of intracellular markers in spleen. Graph (**A**): Interleukin-6 (IL-6) production in stimulated cells from spleen. Graph (**B**): Tumor Necrosis Factor alpha (TNF-α) production in stimulated cells from spleen. Swiss Webster mice were administered one prime dose at week 0 and two booster doses at weeks 2 and 4 via intranasal administration. The spleen was collected at week 11 and analyzed via flow cytometry. There was a significantly higher expression of intracellular cytokines IL-6 and TNF-α than the no treatment group in the spleen. Expression of IL-6 was significantly higher than the TNF-α cytokine. Adjuvanted Zika MP vaccine induced significantly higher IL-6 and TNF-α cytokines than the naïve or no treatment group. (Data represented as Mean ± SEM and a Brown–Forsythe ANOVA test, followed by Dunnett’s multiple comparison test; ns, non-significant, *, *p* < 0.1, **, *p* < 0.01, ***, *p* < 0.001, ****, *p* < 0.0001).

**Table 1 viruses-16-00865-t001:** Characterization of microparticles. Zika vaccine microparticles, Alhydrogel^®^ (Alum) MP, and MPL-A^®^ MPs were characterized for percent of recovery yield, particle size (nm), polydispersity index, zeta potential (mV), and number of particles/mL determined by laser particle counter. Data are represented as Mean ± SD.

	Vaccine MP	Adjuvant MPs	
	Zika Vaccine MP	Alhydrogel MP	MPL-A MP
**Recovery yield %**	90%	87.5%	92%
**Particle size (nm)**	569.0 ± 12.25	1009 ± 24.76	432 ± 8.78
**Polydispersity Index (PDI)**	0.354 ± 0.110	0.438 ± 0.167	0.489 ± 0.167
**Zeta potential (mV)**	−19.42 ± 0.66	−29.78 ± 2.04	−20.67 ± 1.54
**Number of particles/mL**	1180 ± 80	1320 ± 20	1380 ± 50

## Data Availability

Data are available upon request.

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
