# Peer review of "Intranasal Immunization for Zika in a Pre-Clinical Model"

_viruses, 2024, doi:10.3390/v16060865_

Round 1
Reviewer 1 Report
Comments and Suggestions for Authors
The authors designed and developed a microparticlulate ZIKC vaccine based on inactivated whoe zikv virus, through a double emulsion method. Their study showed the vaccine can induce robust humoral responses, for both IgA and IgG. The authors proviede a very valuable attempt in the development of the Zika vaccine.
I have only several minor suggestions and questions:
1) Could the authors provide more information on the purification of inactivated ZIKV and test of purity of ZIKV protein antigens? The BCA can only tell the amount of total proteins. Did the authors do PAGE assay?
2) It appears that their adjuvant has a significant effect on enhancing antibody titers, but its impact on boosting CTL responses and memory is not significant. Would the authors further discuss their adjuvant? Moreover, it would also be interesting to explore the effects of other adjuvants on this MP vaccine in the future.
Reviewer 2 Report
Comments and Suggestions for Authors
Shah et al. evaluated intranasal immunization for ZIKV in a pre-clinical model. They looked at cytotoxicity and different species of antibodies. While this is an interesting study, some experiments lack important controls. The authors failed to look at the effect of neutralizing antibodies. This is a limitation of this study. Detailed review comments can be found below:
Flaviviridae should be italicized.
Ref 1 is not a clinical paper focusing on clinical presentations, diagnosis or treatment of ZIKV. But it is cited to support the statement on clinical symptoms in line 38-39, which may need other references to strengthen the literature support. Moreover, central nervous system infection is missing in the introduction. More references are needed to cite, with the following as an example:
Laboratory diagnosis of CNS infections in children due to emerging and re-emerging neurotropic viruses. Pediatr Res. 2024 Jan;95(2):543-550. doi: 10.1038/s41390-023-02930-6. Epub 2023 Dec 2. PMID: 38042947.
Fig. 1. should add legend rather than just title.
Fig. 3. should add legend rather than just title.
Fig. 4. should add legend rather than just title.
Fig. 7-13: Did the author test cytotoxicity at other concentration of the tested MP or vaccines? Y axis absorbance provides limited information on the antibody quantities. Bar chart is inappropriate and should be changed to line chart.
Fig. 14-15: why did both vaccines provide no difference for CD8+ expression whereas significant difference in CD4+ expression?
Fig. 17: did the authors look at other cytokines/chemokines?
Fig. 8-9: lack other well-established ZIKV vaccine as a positive control
The authors failed to look at the effect of neutralizing antibodies. This is a limitation of this study.
Author Response
Thank you for reviewing our manuscript. Please see the attachment.

Round 2
Reviewer 2 Report
Comments and Suggestions for Authors
The revisions have addressed most of the concerns.
Author Response
Thank you for reviewing our paper.
We have now added neutralization titers under the results section 3.9.